🔓 | **Open Peer Review** | Clinical Microbiology | Research Article

# Quantification of major bacterial pathogens in liver abscesses and matched ruminal and colonic epithelial tissues of feedlot cattle

Mina Abbasi,[1] Alyssa Deters,[1] Harith Salih,[2] Reese A. Wilson,[3] Kasi N. Schneid,[4] Xiaorong Shi,[1] Leigh Ann George,[1] Jianfa Bai,[5] Raghavendra G. Amachawadi,[2] Dale R. Woerner,[3] Kendall L. Samuelson,[4] Ty E. Lawrence,[4] T. G. Nagaraja[1]

**ABSTRACT** Liver abscesses (LA) in cattle are a polymicrobial infection, and the major bacterial pathogens associated are as follows: *Fusobacterium necrophorum* subsp. *necrophorum* (FNN), *F. necrophorum* subsp. *funduliforme* (FNF), *Trueperella pyogenes* (TP), and *Salmonella enterica* (SE). In polymicrobial infections, the contributions of individual species are difficult to assess. We hypothesized that species abundance in abscesses may be indicative of their contributions. Therefore, the objective was to develop a 4-plex quantitative PCR (qPCR) assay to determine prevalence and concentrations of the pathogens in LA ($n = 384$) and matched ruminal (RT; $n = 374$) and colonic epithelial (CT; $n = 256$) tissues. *Fusobacterium necrophorum*, either FNN or FNF, was detected in 85.9% of LA by qPCR, which was slightly lower than the culture-based prevalence (89.1%). Only 16.9% of LA were positive for FNF with no FNN. The concentrations of FNN or FNF were 7 to 7.5 $\log_{10}$ CFU/g. The qPCR assay identified more ($P < 0.01$) samples (29.2%) as positive for TP than the culture method (16.7%). The mean concentration of TP was 5.9 $\log_{10}$ CFU/g. None of the LA was positive for SE by the qPCR assay. The prevalence of FN was greater ($P < 0.01$) in RT than CT (73.2% vs. 16%). The concentrations were in the 4 to 5 $\log_{10}$ CFU/g. The low concentrations of TP suggest that it is unlikely to be the primary etiologic agent. Although SE was detected by culture method, it was not quantifiable, which suggests that the species does not contribute to the development of LA.

**IMPORTANCE** Liver abscesses (LA) in cattle are a polymicrobial infection, and four bacterial pathogens are implicated: *Fusobacterium necrophorum* subsp. *necrophorum* (FNN) and subsp. *funduliforme* (FNF), *Trueperella pyogenes* (TP), and *Salmonella enterica* (SE). In mixed infections, the species abundance may be indicative of their contributions. Our objective was to develop a quantitative PCR assay to determine prevalence and concentrations of the pathogens in LA ($n = 384$) and matched ruminal (RT; $n = 374$) and colonic tissues (CT; $n = 256$). The dominant species in LA was FNN with a mean concentration of 7.0 $\log_{10}$ CFU/g. The subsp. FNF without FNN was prevalent in a small number of LA, with a mean concentration of 7.0 $\log_{10}$CFU. The concentration of TP was 5.0 $\log_{10}$ CFU, which suggests that it is unlikely to be the primary etiologic agent. Although *S. enterica* was prevalent in LA, none was quantifiable, which suggests that it is not contributing to LA.

**KEYWORDS** liver abscesses, qPCR, *Fusobacterium necrophorum*, *Salmonella enterica*, *Trueperella pyogenes*, ruminal and colonic epithelial tissues

**Peer Reviewer** Robert J. Gruninger, Agriculture and Agri-Food Canada Lethbridge Research and Development Centre, Lethbridge, Canada

Address correspondence to T. G. Nagaraja, tnagaraj@vet.k-state.edu.

The authors declare no conflict of interest.

Liver abscesses in feedlot cattle impose an economic burden on the beef cattle industry because of liver condemnations, reduced animal performance, carcass yield and value, and lost time on the slaughter floor (1–4). Historically, liver abscess

development has been linked to ruminal epithelial damage associated with acidosis, which allows bacteria from the rumen to invade the epithelial tissue and proliferate and enter the portal circulation to reach the liver to induce abscesses (5–7). There is some indication that the hindgut may also serve as a source of bacterial pathogens that enter the portal circulation (8).

Both culture-based and culture-independent methods have indicated that liver abscesses are polymicrobial infections dominated by anaerobic bacteria (9–11). *Fusobacterium necrophorum*, a ruminal bacterium, is the most frequently isolated pathogen, followed by *Trueperella pyogenes* and *Salmonella enterica* (9, 10, 12). Both 16S amplicon sequencing-based and metagenomic analyses have affirmed the polymicrobial composition of liver abscesses with Fusobacteriota as the dominant phylum, followed by Bacteroidota and Pseudomonadota(13–17).

*Fusobacterium necrophorum* consists of two subspecies, *necrophorum* and *funduliforme*, which differin morphology, biochemical properties, and virulence (18, 19). Of the two, subsp. *necrophorum* is more virulent because of high leukotoxin production, which enhances its ability to survive and colonize gut epithelial tissues and liver parenchymal tissue to form abscesses (8, 20). The subsp. *funduliforme* is less frequently (approx. 25% of abscesses) associated with liver abscesses (8, 21). In contrast, subsp. *funduliforme* is more commonly prevalent in the rumen, both in contents and epithelial tissues (22, 23).

In polymicrobial infections, it is difficult to assess the contributions of individual bacterial species. We hypothesized that bacterial species abundance in purulent materials of liver abscesses may be the best indicator of their contributions as etiologic agents. Culture-based methods are more likely to identify the dominant species present in such infections but do not provide quantitative data unless selective media are available for targeting specific species, which are time-sensitive and labor-intensive. The DNA sequence-based analyses, particularly 16S amplicon sequence-based, can provide quantitative data based on DNA concentrations or read counts, but do not provide species-level resolution because of the conserved nature of the gene and the number of copies of 16S rRNA genes (24). Real-time quantitative PCR (qPCR) provides a specific method for species quantification. Quantitative PCR assays have been developed individually for *F. necrophorum* at the species and subspecies levels (25–29). Deters et al. (22) reported a qPCR assay for the detection and quantification of the two subspecies of *F. necrophorum* by targeting the promoter region of the leukotoxin gene, *lktA*, and to differentiate the two subspecies from another species of *Fusobacterium*, called *F. varium*, that is prevalent in ruminal and colonic epithelial tissues and sometimes in liver abscesses (22). Similarly, the *plo* gene, which encodes for pyolysin, is targeted for detection and quantification of *T. pyogenes* by conventional PCR (30, 31) or qPCR (32) assay, respectively. For *S. enterica*, the *invA*, which encodes an invasion protein, is commonly targeted for detection and quantification (33, 34). The concentrations of the two subspecies of *F. necrophorum* have been reported to be in the range of 6 to 7 $\log_{10}$ CFU/g of purulent materials of liver abscesses, but the concentrations of *T. pyogenes* and *S. enterica* have not been determined. Quantification of *T. pyogenes* and *S. enterica* in purulent materials of liver abscesses may provide an assessment of their contributions to liver abscess development. The objective of this study was to develop and validate a 4-plex qPCR assay for the detection and quantification of the four major liver abscess pathogens, *F. necrophorum* subsp. *necrophorum* and subsp. *funduliforme*, *T. pyogenes*, and *S. enterica* in liver abscesses and matched ruminal epithelial and colonic epithelial tissues of feedlot cattle.

## MATERIALS AND METHODS

### Design of primers and probes

The genes or genetic regions of the four pathogens that were targeted for the assay included the promoter region of the leukotoxin gene, *lktA*, for the two *F. necrophorum* subspecies (22), the *plo* gene for *T. pyogenes* (31), and the *invA* gene for *S. enterica*

(34). The primer designs were based on evaluations of sequences available in Gen-Bank (https://www.ncbi.nlm.nih.gov/genbank/) at the time, with sequence alignment performed using ClustalX version 2.1 and viewed in BioEdit version 7.1.3.0, for primer and probe selections. Primer and probe candidates with the most matched target sequences that did not match closely related nontarget sequences were chosen for further analyses (Table 1). The selected primers and probes were synthesized by and obtained from Integrated DNA Technologies (Coralville, IA) (Table 1).

## Optimization of the assay

Four strains each of *F. necrophorum* subsp. *necrophorum* (2024-2-1, 2024-2-6, 2024-2-11, and 2024-2-16) and subsp. *funduliforme* (NDSU20, NDSU21, NDSU23, and NDSU26), six strains of *T. pyogenes* (2024-1-132, 2024-1-163, 2024-1-164, 2024-1-183, 2024-1-223, and MS2011), and five *S. enterica* strains (2024-1-121, 2024-1-122, 2024-1-125, 14028, and 700408), previously isolated from liver abscesses of cattle, were used for qPCR optimization. Each isolate was streaked onto blood agar plates (BAP) and incubated overnight at 37°C. Three to five colonies were collected from each plate, suspended in 1 mL of nuclease-free water, boiled for 10 minutes, and centrifuged at $9,300 \times g$ for 5 minutes. The lysate supernatant was used as the DNA template for the qPCR assay to test for specificity.

The qPCR reactions were performed on a BioRad CFX96 Real-Time PCR Detection System (BioRad, Hercules, CA) in a 20 µL reaction volume. Each reaction contained 10 µL of iQ Multiplex Powermix (2X; BioRad, Cat#: 172-5849), 1 µL of an 8-primer mix (two subspecies of FN, *S. enterica*, *T. pyogenes*, each at 5 pM/µL), 1 µL of subsp. *necrophorum* probe (10 pM/µL, FAM), 1 µL of subsp. *funduliforme* probe (10 pM/µL, HEX), 1 µL of *S. enterica* probe (5 pM/µL, Texas Red), 1 µL of *T. pyogenes probe* (10 pM/µL, Cy5), 3 µL of nuclease-free water, and 2 µL of extracted DNA. The thermal cycling protocol included an initial denaturation at 95°C for 10 minutes, followed by 45 cycles of 95°C for 15 seconds and 60°C for 40 seconds. Each run included a negative control (nuclease-free water) and positive controls (DNA from pure cultures of targeted species and subspecies) to ensure accuracy and reproducibility.

## Assay specificity

The specificity of the assay was evaluated using 172 bacterial strains representing 28 species, including target species and nontarget species, encompassing both Gram-positive and Gram-negative bacteria. Among the tested isolates, 10 strains of other *Fusobacterium* species—*F. equinum* (1), *F. gastrosuis* (1), *F. nucleatum* (1), *F. naviforme,* H335 (1), *F. russiii* (1), *F. ulcerans* (1), *F. varium* (3), and *F. gonidioformans* (1)—were included, along with five strains of *F. necrophorum* subsp. *necrophorum* (2024-2-1, 2024-2-6, 2024-2-11, 2024-2-16, and 8L1); six strains of *F. necrophorum* subsp. *funduliforme* (NDSU20, NDSU21, NDSU23, NDSU26, FN-H3, and FN-H4); 12 strains *of T. pyogenes* (2024-1-132, 2024-1-163, 2024-1-164, 2024-1-183, 2024-1-223, MS 2011, 2022-4-251, 2022-4- 252, 2023-3-19, 2023-3-20, 2023-3-22, and 2023-3-23) isolated from liver abscesses, ruminal epithelial tissues, and lung tissues; 66 strains of *Salmonella enterica* serotypes: *S*. Typhimurium (3), *S*. Agona (1), *S*. Albany (1), *S*. Altona (1), *S*. Anatum (3), *S*. Bareilly (1), *S*. Bere (1), *S*. Bergen (1), *S*. Brandenburg (1), *S*. Bredeney (1), *S*. Cerro (1), *S*. Cubana (1), *S*. Derby (1), *S*. Dublin (1), *S*. Enterica Endentidis (2), *S*. Fresno (1), *S*. Gallinarum (1), *S*. Gaminara (1), *S*. Give (1), *S*. Havana (1), *S*. Infantis (1), *S*. Kedougou (1), *S*. Kentucky (1), *S*. Kiambu (1), *S*. Lille (1), *S*. Lubbock (1), *S*. Mbandaka (1), *S*. Meleagridis (1), *S*. Montevideo (1), *S*. Muenchen (1), *S*. Muenster (2), *S*. Newport (1), *S*. Norwich (1), *S*. Oranienburg (1), *S*. Orion (2), *S*. Reading (1), *S*. Rough (9), *S*. Saintpaul (1), *S*. Sandiego (1), *S*. Schwarzengrund (1), *S*. Senftenberg (1), *S*. Sundsvall (1), *S*. Thompson (1), *S*. Uganda (2), and six un-typable strains of *Salmonella* (6); 16 *Escherichia coli* strains, which included serogroups of O26 (3), O45 (4), O103 (1), O111 (2), O121 (1), O145 (1), O157 (1), O104 (1), O74 (1), and the ATCC strain 25922; and 21 *Enterococcus* strains: *E. casseliflavis* (2), *E. faecalis* (4), *E. faecium* (3), *E. gallinarum* (10), *E. hirae* (1), and *E. mundtii* (1). Additionally,

**TABLE 1** Bacterial species and subspecies, targeted genes, primer and probe sequences, and amplicon sizes in the 4-plex quantitative PCR assay

| Target | Gene target | Primer sequences (5′–3′)[1] | Length, bp |
|---|---|---|---|
| *Fusobacterium necrophorum* subsp. *necrophorum* | *lkt*A | Forward: GCTTTGGAAGAAGCCAAACA | 93 |
| | | Reverse: ATGCTTCCATTCGGATTCA | |
| | | Probe: FAM-TGGAATCATTCCAGTAGATGGAAAAG-ZEN/3′IB | |
| *Fusobacterium necrophorum* subsp. *funduliforme* | *lkt*A | Forward: AAAGACGCTCAAAATAGCAAAGTT | 80 |
| | | Reverse: TTTGGATTCAACGGAATCTTG | |
| | | Probe: HEX-TTGTTCCACAACAGGATGGGAGTA-ZEN/3′IB | |
| *Trueperella pyogenes* | *plo* | Forward: TCGGATTTGAAAAGGTCTCAG | 96 |
| | | Reverse: AAATCTGTTTGAAGGAAGCGATA | |
| | | Probe: Cy5-CGTGGACTTCGATGCAATTCA-3′IB | |
| *Salmonella enterica* | *inv*A | Forward: CGTGTT TCCGTG CGTAATA | 138 |
| | | Reverse: GCCATTGGCGAATTTATG | |
| | | Probe: TexasRed-ATT ATG GAA GCG CTC GCA TT-3′IB | |

other species included *Actinobacillus pleuropneumoniae* (1), *Bacillus cereus* (1), *Bacillus subtilis* (1), *Bordetella bronchiseptica* (1), *Campylobacter jejuni* (1), *Campylobacter coli* (1), *Campylobacter fetus* (1), *Campylobacter lari* (1), *Campylobacter upsaliensis* (1), *Campylobacter hyointestinalis* (1), *Corynebacterium pseudotuberculosis* (1), *Enterobacter aerogenes* (2), *Ersipelothrix rhusiopathiae* (1), *Histophilus somni* (1), *Klebsiella pneumoniae* (1), *Listeria monocytogenes* (1), *Mannheimia haemolytica* (1), *Moraxella bovoculi* (1), *Morganella morganii* (2), *Mycobacterium fortuitum* (1), *Pasteurella multocida* (1), *Proteus mirabilis* (1), *P. vulgaris* (1), *P. rettgeri* (1), *Pseudomonas aeruginosa* (2), *Rhodococcus equi* (1), *Serratia marcescens* (2), *Staphylococcus aureus* (1), *Streptococcus agalactiae* (1), *Streptococcus equi* subsp. *zooepidemicus* (1), *Streptococcus pyogenes* (1), and *Streptococcus suis* (1). The DNA extract from the strains was prepared as described above.

## Assay sensitivity and standard curve analysis using pure cultures

The sensitivity of the assay was determined using pure cultures of the targeted species and subspecies (strain): *F. necrophorum* subsp. *necrophorum* (8L1) and *funduliforme* (2023-3-506), *T. pyogenes* (2022-4-252), and *S. enterica* (2016-13-36), all previously isolated from liver abscesses of cattle. Serial dilutions of cultures grown to the late logarithmic phase were used to determine bacterial concentrations, expressed as colony-forming units per mL (CFU/mL), by the spread plate method. The DNA was extracted from the cultures as described above, serially diluted, and assayed by qPCR to assess the lower limit of detection (LLOD).

*Fusobacterium* strains, stored at −80°C in pre-reduced, anaerobically sterilized (PRAS) brain-heart infusion (BHI) agar slants, were cultured on blood agar (Remel Inc., Lenexa, KS) under anaerobic conditions at 37°C for 48 hours. *Salmonella* and *Trueperella* strains, stored at −80°C, were streaked onto blood agar. Single colonies from each strain were transferred into 10 mL of BHI broth and incubated overnight at 37°C. A 100 µL aliquot of the overnight culture was then inoculated into 10 mL of fresh BHI broth and incubated at 37°C until the absorbance at 600 nm reached 0.4 (for *F. necrophorum* subspecies), 0.34 (for *T. pyogenes*), and 0.58 (for *S. enterica*) (~3.5 hours, corresponding to $10^7$–$10^8$ CFU/mL). Ten-fold serial dilutions were prepared in BHI broth, and 100 µL from $10^{-5}$, $10^{-6}$, and $10^{-7}$ dilutions were spread onto BAP to determine viable bacterial cell concentrations. From each dilution, 1 mL of culture was boiled for 10 minutes, centrifuged at 9,300 × *g* for 5 minutes, and the supernatant was used as the qPCR template. All PCR reactions were performed in triplicate, and each standard curve was replicated for each strain. Correlation coefficients and PCR amplification efficiencies were analyzed using CFX Manager software (BioRad, Hercules, CA).

## Application of the 4-plex qPCR assay to determine prevalence and concentrations of bacterial pathogens in liver abscesses, ruminal epithelial tissues, and colonic epithelial tissues

Liver abscesses, ruminal epithelial tissues, and colonic epithelial tissues from beef cattle were collected at slaughter from three independent feedlot studies, two conducted in 2022 and one in 2023. Two studies included all three tissue types, while one included only liver abscesses and ruminal epithelial tissues. Study one focused on determining the prevalence of major and minor pathogens in liver abscesses and matched ruminal and colonic epithelial tissues from feedlot cattle fed a finishing diet with no in-feed tylosin. Samples were obtained from 96 cattle that originated from 15 Midwestern US feedlots, with one to 19 animals sampled per feedlot. Tissue collections occurred at a commercial slaughter facility in the southern United States over a seven-month period (June to December 2022). A total of 96 liver abscess samples and 96 matched ruminal epithelial and colonic tissue samples were collected. The second study assessed the impact of in-feed tylosin and an antibiotic alternative, *Saccharomyces cerevisiae* CNCM I-1077 (yeast) combined with calcium clinoptilolite zeolite, on liver abscess incidence in beef-on-dairy heifers. Heifers were fed a finishing diet for an average of 228 days (range: 205 to 262 days) before harvest across seven dates (Oct 7, 2022, to Feb 28, 2023). Three treatment groups were included: (1) negative control (no intervention), (2) tylosin group (68 mg tylosin/heifer/day), and (3) yeast + zeolite group (0.5 g *S. cerevisiae*/heifer/day + 1.2% dietary dry matter zeolite). A total of 111 liver abscess samples and 102 corresponding ruminal epithelial tissue samples were collected. The study population, treatment allocation, diet, management, liver abscess incidence, and bacteriological findings by culture methods have been previously described (23, 35, 36). The third study examined the effects of dietary starch concentrations and feeding regimen on growth performance, carcass traits, and liver abscess prevalence in feedlot steers. The study design was a $2 \times 2$ factorial with steers assigned to one of four treatment groups: low starch (49.0% of the diet) with consistent or erratic feeding regimen and high starch (64.4% of the diet) with consistent or erratic feeding regimen. None of the diets contained tylosin, but monensin was included in the diet. Details on study design, animal allocation, dietary composition, and incidence of liver abscesses have been described (37). At harvest, a total of 177 liver abscesses and 176 matched ruminal epithelial and 160 matched colonic epithelial tissue samples were collected from a subset of steers. All samples were stored on ice and shipped overnight to the Anaerobe Laboratory in the College of Veterinary Medicine at Kansas State University for bacteriological analyses.

The culture method data in the tables include isolations before enrichment and total isolations, which combine those before and after enrichment. We have compared prevalence by qPCR with culture method before enrichment because both were done with the pre-enrichment samples. Because post-enrichment samples were not saved, qPCR assay could not be performed. The culture-based data were pooled from three individual studies and were included to compare with the prevalence determined by qPCR (25, 37, 38).

### Sample processing

Approximately 5 g of purulent and pieces of capsular materials of liver abscesses, ruminal papillae, or colonic mucosal layers were suspended in 45 mL of sterile PBS, and the exact sample weight was recorded. Sample suspensions were homogenized for 1 minute in a Nutribullet Pro Blender (Nutribullet, Los Angeles, CA). Aliquots of the homogenates were stored at −80°C.

### DNA extraction

Sample homogenates were allowed to thaw in the refrigerator. The DNA was extracted from the homogenates using the MagMax-96 DNA Multi-Sample Kit (Applied Biosystems, Waltham, MA) according to the manufacturer's protocol. Briefly, 100 µL of tissue

homogenates were mixed with 100 µL of Multi-Sample DNA Lysis Buffer, incubated at room temperature for 10 minutes, and then processed using a KingFisher Flex machine (Thermo Fisher Scientific, Waltham, Massachusetts).

## Statistical analysis

Data were analyzed in R statistical software (version 4.4.2, R Core Team, Vienna, Austria). The prevalence of the four bacterial species (*F. necrophorum* subsp. *necrophorum*, *F. necrophorum* subsp. *funduliforme*, *T. pyogenes,* and *S. enterica*), as detected by qPCR or culture methods of pre-enrichment samples, in liver abscesses, ruminal epithelial tissues, and colonic epithelial tissues was analyzed using a generalized linear mixed-effects model ('glmer' function) from the lmerTest package (39). Only pre-enrichment culture prevalence was statistically compared to the qPCR results because both used the same sample. The total prevalence by culture method included prevalence before and after enrichment combined. Pairwise comparisons were conducted using Tukey *P*-value adjustment ('emmeans' function) from the multcomp package (40). The concentrations of the four bacterial species, as determined by qPCR, were log (base-10)-transformed and analyzed using a linear mixed-effects model (lmer function) from the lmerTest package. For all analyses, detection methods or tissue types were set as a fixed effect, and the feedlot study was included as a random effect. Pairwise comparisons of concentration also used a Tukey adjustment. Differences were considered significant at $P \leq 0.05$, and meaningful tendencies were discussed when $P > 0.05$ and $P \leq 0.10$.

## RESULTS

All strains of the four targeted organisms—two subspecies of *F. necrophorum*, *T. pyogenes, and S. enterica*—yielded positive amplifications with the designed primers and probes. None of the other Gram-positive or Gram-negative bacterial species yielded any amplifications. The sensitivities of the assay indicated that the lower limit of detection (LLOD) based on pure cultures was $4.89 \times 10^2$ CFU/mL for *F. necrophorum* subsp. *necrophorum*, $1.9 \times 10^3$ CFU/mL for subsp. *funduliforme*, $2.2 \times 10^3$ CFU/mL for *T. pyogenes*, and $8.6 \times 10^2$ CFU/mL for *S. enterica*. The qPCR assay demonstrated strong linearity and amplicon efficiency across the four targets. The amplification efficiencies and correlation coefficients ($R^2$) were 103.1% ($R^2 = 0.998$) for *F. necrophorum* subsp. *necrophorum*, 97.4% ($R^2 = 0.998$) for *F. necrophorum* subsp. *funduliforme*, 97.2% ($R^2 = 0.998$) for *S. enterica*, and 98.8% ($R^2 = 0.999$) for *T. pyogenes*. The values were within the acceptable range for qPCR, supporting the assay's sensitivity and reliability for multiplex detection.

A total of 384 liver abscesses, 374 ruminal epithelial tissues, and 256 colonic epithelial tissues were used in the study. The lower number of ruminal and colonic epithelial tissue samples relative to liver abscesses was due to the absence of colonic tissue collection in the second feedlot study. In a few instances, matched ruminal and colonic epithelial tissues could not be recovered for every abscessed liver collected. To compare qPCR-assay prevalence with the culture-based method, prevalence data of the two subspecies of *F. necrophorum*, *T. pyogenes*, and *S. enterica* in samples before enrichment and total prevalence, including post-enrichment prevalence, are included (23, 41, 42).

## Prevalence and concentrations of *F. necrophorum*, *T. pyogenes*, and *S. enterica* in liver abscesses

Of the 384 liver abscess samples analyzed, *F. necrophorum*, either subsp. *necrophorum* or *funduliforme*, was detected in 330 of liver abscess samples by qPCR, which was slightly less than the detection rate by the culture-based method before enrichment (342/384), but the difference in the prevalence between the two methods was not significant (85.9% vs. 89.1%; *P* = 0.17; Table 2). Only a small proportion (21.4%) of samples were positive for both subspecies by qPCR, which was similar to the culture method (19.3%). In all the samples that were positive for either or both subspecies, the mean concentrations were approximately 7 $\log_{10}$ CFU/g of purulent material. The qPCR assay identified more

TABLE 2 Prevalence and concentrations of *Fusobacterium necrophorum*, *Trueperella pyogenes*, and *Salmonella enterica*, alone or in combination, in liver abscesses of feedlot cattle, based on 4-plex quantitative PCR (qPCR) and culture methods[b]

| Bacterial species | Prevalence and concentration | No. of samples positive (%) | | | P value (qPCR vs. before enrichment) |
| --- | --- | --- | --- | --- | --- |
| | | qPCR | Culture method | | |
| | | | Before enrichment | Total[a] | |
| No. of samples | | 384 | 384 | 384 | NA[c] |
| *F. necrophorum* subsp. *necrophorum* or subsp. *funduliforme* | Prevalence | 330 (85.9) | 342 (89.1) | 351 (91.4) | 0.17 |
| | Log$_{10}$ CFU/g | 7.26 or 7.06 | NA | NA | NA |
| *F. necrophorum* subsp. *necrophorum* and subsp. *funduliforme* | Prevalence | 82 (21.4) | 74 (19.3) | 94 (24.5) | 0.47 |
| | Log$_{10}$ CFU/g | 7.01 and 6.91 | NA | NA | NA |
| *T. pyogenes* | Prevalence | 112 (29.2) | 64 (16.7) | 64 (16.7) | <0.01 |
| | Log$_{10}$ CFU/g | 5.89 | NA | NA | NA |
| *S. enterica* | Prevalence | 0 | 12 (3.1) | 76 (19.8) | <0.01 |
| | Log$_{10}$ CFU/g | ND | NA | NA | NA |
| *F. necrophorum* + *T. pyogenes* | Prevalence | 111 (28.9) | 63 (16.4) | 63 (16.4) | <0.01 |
| | Log$_{10}$ CFU/g | 7.37 and 5.89 | NA | NA | NA |
| *F. necrophorum* + *S. enterica* | Prevalence | 0 | 11 (2.9) | 72 (18.8) | <0.01 |
| | Log$_{10}$ CFU/g | ND | NA | NA | NA |
| *F. necrophorum* + *T. pyogenes* + *S. enterica* | Prevalence | 0 | 2 (0.5) | 5 (1.3) | 0.16 |
| | Log$_{10}$ CFU/g | ND | NA | NA | NA |

[a]Total includes prevalence before and after enrichment for *F. necrophorum* and *S. enterica*.
[b]ND, not detected.
[c]NA, not applicable.

samples (112/384; 29.2%) as positive for *T. pyogenes* than the culture method (16.7%; *P* < 0.01). The prevalence of *T. pyogenes* before enrichment and the total prevalence were the same because no enrichment method has been developed for *T. pyogenes*. The mean concentration of *T. pyogenes* was about one log lower (5.89 log$_{10}$ CFU/g) than the two subspecies of *F. necrophorum*. None of the liver abscess samples was positive for *S. enterica* by the qPCR assay, and only a small number of samples (12/384; 3.1%) were positive before enrichment by the culture method. However, a higher proportion of samples (19.8%) was positive when samples were subjected to an enrichment step to detect *Salmonella* (Table 2).

The combination of *F. necrophorum* and *T. pyogenes* was prevalent in 28.9% (111/384) of liver abscesses, and only one sample contained *T. pyogenes* without *F. necrophorum*. The qPCR assay detected more samples as positive for both species than the culture method (28.9% vs. 16.4%; *P* < 0.01). The concentration of *F. necrophorum* when present with *T. pyogenes* was similar to the concentration in the absence of *T. pyogenes*. Because all the samples were negative for *S. enterica* by the qPCR assay, none of the samples was positive for the combinations of *F. necrophorum* and *S. enterica* or *F. necrophorum*, *T. pyogenes*, and *S. enterica*. However, based on the culture method, a small number of samples, either before enrichment (11/384; 2.9%) or after enrichment (72/384; 18.8%), were positive for both *F. necrophorum* and *S. enterica*. Only five samples contained all three bacterial species, which were detected by the culture method (Table 2).

Among the two subspecies, more samples were positive for subsp. *necrophorum* (69%) than *funduliforme* (38.3%; Table 3). Of the samples positive for subsp. *necrophorum*, 47.7% were positive for subsp. *necrophorum* only, with no *funduliforme* present, which was similar to the prevalence by the culture method (50.3%). In contrast, only a small proportion of liver abscess samples was positive for *funduliforme* only with no subsp. *necrophorum* (16.9% and 19.5% by the qPCR and culture method, respectively) (Table 3). The concentrations of the two subspecies alone were similar to the concentrations when both were present. The prevalence of the two subspecies in association with *T. pyogenes* was similar, and the concentration of *F. necrophorum* was not affected by the presence of *T. pyogenes*. Because all the samples were negative for *S. enterica* by the qPCR method, none of the samples tested positive for combinations of *F. necrophorum* and *S. enterica* or *F. necrophorum*, *T. pyogenes*, and *S. enterica*. However, based on the total prevalence

**TABLE 3** Prevalence and concentrations of the two subspecies of *Fusobacterium necrophorum*, alone or in combination with *Trueperella pyogenes* and/or *Salmonella enterica* in liver abscesses of feedlot cattle, based on 4-plex quantitative PCR (qPCR) and culture methods[b]

| Bacterial species | Prevalence and concentration | No. of samples positive (%) | | | P value (qPCR vs. before enrichment) |
| | | qPCR | Culture method | | |
| | | | Before enrichment | Total[a] | |
| --- | --- | --- | --- | --- | --- |
| No. of samples tested | | 384 | 384 | 384 | NA[c] |
| *F. necrophorum* subsp. *necrophorum* | | | | | |
| Alone (without subsp. *funduliforme*) | Prevalence | 183 (47.7) | 193 (50.3) | 181 (47.1) | 0.45 |
| | Log$_{10}$ CFU/g | 7.37 | NA | NA | |
| Total (with or without subsp. *funduliforme*) | Prevalence | 265 (69.0) | 267 (69.5) | 275 (71.6) | 0.94 |
| | Log$_{10}$ CFU/g | 7.26 | NA | NA | NA |
| Plus *T. pyogenes* | Prevalence | 74 (19.3) | 40 (10.4) | 40 (10.4) | <0.01 |
| | Log$_{10}$ CFU/g | 7.31 + 5.99 | NA | NA | |
| Plus *S. enterica* | Prevalence | ND | 9 (2.3) | 57 (14.8) | <0.01 |
| | Log$_{10}$ CFU/g | ND | NA | NA | NA |
| Plus *T. pyogenes* and *S. enterica* | Prevalence | ND | 1 (0.3) | 1 (0.3) | 0.31 |
| | Log$_{10}$ CFU/g | ND | NA | NA | NA |
| *F. necrophorum* subsp. *funduliforme* | | | | | |
| Alone (without subsp. *necrophorum*) | Prevalence | 65 (16.9) | 75 (19.5) | 76 (19.8) | 0.35 |
| | Log$_{10}$ CFU/g | 7.24 | NA | NA | NA |
| Total (with or without subsp. *necrophorum*) | Prevalence | 147 (38.3) | 149 (38.8) | 170 (44.3) | 0.94 |
| | Log$_{10}$ CFU/g | 7.06 | NA | NA | NA |
| Plus *T. pyogenes* | Prevalence | 75 (19.5) | 38 (9.9) | 38 (9.9) | <0.01 |
| | Log$_{10}$ CFU/g | 7.43 + 5.76 | NA | NA | NA |
| Plus *S. enterica* | Prevalence | ND | 5 (1.3) | 44 (11.5) | 0.02 |
| | Log$_{10}$ CFU/g | ND | NA | NA | NA |
| Plus *T. pyogenes* and *S. enterica* | Prevalence | ND | 2 (0.5) | 2 (0.5) | 0.16 |
| | Log$_{10}$ CFU/g | ND | NA | NA | NA |

[a]Total includes prevalence before and after enrichment for *F. necrophorum* and *S. enterica*.
[b]ND, not detected.
[c]NA, not applicable.

of *Salmonella* by culture method, which included post-enrichment isolation, a small proportion of liver abscesses (14.8% and 11.5% for the subsp. *necrophorum* and subsp. *funduliforme*, respectively) were positive for both. Only one or two samples contained all three bacterial species (Table 3).

## Prevalence and concentrations of *F. necrophorum*, *T. pyogenes*, and *S. enterica* in ruminal and colonic epithelial tissues

The qPCR assay detected more ruminal epithelial tissues as positive for either of the two subspecies of *F. necrophorum* (76.7% vs. 16%; $P < 0.01$) or for both subspecies together (35.8% vs. 1.2%; $P < 0.01$) compared to colonic epithelial tissues (Table 4). Interestingly, the prevalence determined by the qPCR method for the two subspecies was greater ($P < 0.01$) in ruminal epithelial tissues, but lower ($P < 0.01$) in colonic epithelial tissues than the culture method-based prevalence. The prevalence of *T. pyogenes* by the qPCR assay (19.0% vs. 5.5%) and the culture method (21.7% vs. 4.7%) was higher ($P < 0.01$) in ruminal epithelial tissues than in colonic epithelial tissues. Similar to liver abscess samples, none of the ruminal epithelial tissues tested positive for *S. enterica* by qPCR assay, and only a small number (7/374; 1.9%) were positive by the culture method. However, after enrichment, the prevalence was 26.7% by the culture method. In contrast, a small number of colonic epithelial tissues (17/256; 6.6%) were positive by the qPCR assay. However, only six of the 17 colonic epithelial tissue samples yielded *S. enterica* in pure culture. The total prevalence, which included isolation after enrichment, was 43.8% in colonic tissues compared to 26.7% of ruminal epithelial tissue ($P = 0.01$; Table 4). The prevalence of *T. pyogenes*, in combination with either of the two subspecies

**TABLE 4** Prevalence of *Fusobacterium necrophorum*, *Trueperella pyogenes*, and *Salmonella enterica* in ruminal and colonic epithelial tissues of feedlot cattle, based on 4-plex quantitative PCR (qPCR) assay and culture-based methods

| Bacterial species | Ruminal epithelial tissue | | | P value | Colonic epithelial tissue | | | P value (qPCR | Ruminal vs. colonic | |
|---|---|---|---|---|---|---|---|---|---|---|
| | qPCR | Culture method | | (qPCR vs. before | qPCR | Culture method | | vs. before | | |
| | | Before enrichment | Total[a] | enrichment) | | Before enrichment | Total[a] | enrichment) | P value (qPCR) | P value (Culture) |
| No of samples tested | 374 | 374 | 374 | NA[b] | 256 | 256 | 256 | NA | NA | NA |
| *F. necrophorum* subsp. *necrophorum* or subsp. *funduliforme* | 287 (76.7) | 190 (50.8) | 239 (63.9) | <0.01 | 41 (16.0) | 68 (26.6) | 88 (34.4) | <0.01 | <0.01 | <0.01 |
| *F. necrophorum* subsp. *necrophorum* and subsp. *funduliforme* | 134 (35.8) | 27 (7.2) | 46 (12.3) | <0.01 | 3 (1.2) | 10 (3.9) | 12 (4.7) | 0.05 | <0.01 | 0.02 |
| *T. pyogenes* | 71 (19.0) | 81 (21.7) | 81 (21.7) | 0.21 | 14 (5.5) | 12 (4.7) | 12 (4.7) | 0.69 | <0.01 | <0.01 |
| *S. enterica* | 0 | 7 (1.9) | 100 (26.7) | <0.01 | 17 (6.6) | 6 (2.3) | 112 (43.8) | 0.01 | < 0.01 | 0.84 |
| *F. necrophorum* + *T. pyogenes* | 64 (17.1) | 52 (13.9) | 63 (16.8) | 0.29 | 11 (4.3) | 0 | 3 (1.2) | <0.01 | <0.01 | <0.01 |
| *F. necrophorum* + *S. enterica* | 0 | 1 (0.3) | 72 (19.3) | 0.31 | 3 (1.2) | 1 (0.4) | 34 (13.3) | 0.31 | 0.04 | 0.79 |
| *F. necrophorum* + *T. pyogenes* + *S. enterica* | 0 | 0 | 18 (4.8) | NA | 1 (0.4) | 0 | 0 | NA | NA | NA |

[a]Total includes prevalence before and after enrichment for *F. necrophorum* and *S. enterica*.
[b]NA, not applicable.

of *F. necrophorum*, was similar between the two methods of detection ($P = 0.21$), and the prevalence by both methods was greater in ruminal epithelial tissues than the colonic epithelial tissues (17.1% vs. 4.3% and 13.9% vs. 0% for qPCR and culture method, respectively) (Table 4). Based on the qPCR assay, none of the ruminal epithelial tissues was positive for combinations of *F. necrophorum* and *S. enterica*, but three of the 256 (1.2%) colonic epithelial tissues contained both species. None of the ruminal epithelial tissues was positive for all three species based on the qPCR assay; however, only one colonic epithelial tissue sample had all three species by qPCR (Table 4).

The total prevalence of subsp. *necrophorum*, either alone or with subsp. *funduliforme*, based on the qPCR method was higher in ruminal epithelial tissue than colonic epithelial tissue (39.3% vs. 8.2%; $P < 0.01$), but the prevalence by culture method was not different (11.8% vs. 9.8%; $P = 0.52$). Only a small number of ruminal epithelial tissues (13/374; 3.5%) were positive for subsp. *necrophorum* only, without the subsp. *funduliforme*, and was lower ($P = 0.04$) than the prevalence in colonic epithelial tissues (18/256; 7.0%). In contrast to subsp. *necrophorum*, the total prevalence of subsp. *funduliforme* in ruminal tissue was high (73.3%), and in almost half of those samples (37.4%), it was detected alone, without subsp. *necrophorum* (Table 5). Similar to subsp. *necrophorum*, the qPCR assay detected more ruminal epithelial samples as positive for subsp. *funduliforme* than the culture method before enrichment (73.3% vs. 46.3%; $P < 0.01$) (Table 6). In the colonic epithelial tissue, the culture method detected greater ($P < 0.01$) number of samples as positive for subsp. *funduliforme* than the qPCR assay (20.7% vs. 9.0%).

The concentrations of the two subspecies of *F. necrophorum* when present alone were similar to the concentrations when present in combination. The concentrations of the two subspecies, individually or together, were greater ($P < 0.01$) in ruminal epithelial tissues than in colonic epithelial tissues (Table 6). The concentrations of subsp. *necrophorum*, without subsp. *funduliforme* in ruminal epithelial tissues, were not different ($\log_{10}$ 4.95 vs. 4.92; $P = 0.79$) from the concentration in colonic epithelial tissues. However, the concentrations of subsp. *funduliforme* with or without subsp. *necrophorum* were greater ($P < 0.01$) in ruminal epithelial tissues than in colonic epithelial tissues ($\log_{10}$ 5.19 vs. 4.78; Table 6). The mean concentration of *T. pyogenes* in ruminal epithelial tissue was higher than in colonic epithelial tissues ($\log_{10}$ 5.00 vs. 4.66; $P = 0.02$). The concentrations

**TABLE 5** Prevalence of *Fusobacterium necrophorum* subsp. *necrophorum* and subsp. *funduliforme* in ruminal and colonic epithelial tissues of feedlot cattle, based on quantitative 4-plex quantitative PCR (qPCR) assay and culture-based methods[b]

| Bacterial species | Ruminal epithelial tissue | | | P value (qPCR vs. before enrichment) | Colonic epithelial tissue | | | P value (qPCR vs. before enrichment) | Ruminal vs. colonic | |
| --- | --- | --- | --- | --- | --- | --- | --- | --- | --- | --- |
| | qPCR | Culture method | | | qPCR | Culture method | | | P value (qPCR) | P value (Culture) |
| | | Before enrichment | Total[a] | | | Before enrichment | Total[a] | | | |
| No. of samples tested | 374 | 374 | 374 | | 256 | 256 | 256 | NA[c] | NA | NA |
| *F. necrophorum* subsp. *necrophorum* | | | | | | | | | | |
| Alone (without subsp. *funduliforme*) | 13 (3.5) | 17 (4.5) | 12 (3.2) | 0.45 | 18 (7.0) | 15 (5.9) | 14 (5.5) | 0.59 | 0.04 | 0.45 |
| Total[a] (with or without subsp. *funduliforme*) | 147 (39.3) | 44 (11.8) | 58 (15.5) | <0.01 | 21 (8.2) | 25 (9.8) | 26 (10.2) | 0.65 | <0.01 | 0.52 |
| *F. necrophorum* subsp. *funduliforme* | | | | | | | | | | |
| Alone (without subsp. *necrophorum*) | 140 (37.4) | 146 (39.0) | 181 (48.4) | 0.13 | 20 (7.8) | 43 (16.8) | 62 (24.2) | <0.01 | <0.01 | <0.01 |
| Total (with or without subsp. *necrophorum*) | 274 (73.3) | 173 (46.3) | 227 (60.7) | <0.01 | 23 (9.0) | 53 (20.7) | 74 (28.9) | <0.01 | <0.01 | <0.01 |

[a]Total includes prevalence before and after enrichment for *F. necrophorum* and *S. enterica*.
[b]ND, not detected.
[c]NA, not applicable.

of both *F. necrophorum* and *T. pyogenes* when present together were higher in ruminal epithelial tissues than in colonic epithelial tissues. None of the ruminal epithelial tissues had quantifiable concentration of *S. enterica*; however, a few (17/256) colonic epithelial tissues contained mean concentration of *S. enterica* at $\log_{10}$ 4.91 CFU/g (Table 6).

**TABLE 6** Concentrations of *Fusobacterium necrophorum*, *Trueperella pyogenes*, and *Salmonella enterica*, individually or in combination to ruminal and colonic epithelial tissues of feedlot cattle, based on 4-plex quantitative PCR (qPCR) assay and culture-based method[a]

| Bacterial species | Ruminal epithelial tissue (n = 374) | Colonic epithelial tissue (n = 256) | SEM | P value (rumen vs. colon) |
| --- | --- | --- | --- | --- |
| *F. necrophorum* subsp. *necrophorum* or subsp. *funduliforme* | 5.25 or 5.39 | 4.94 or 4.82 | 0.10 | <0.01 |
| *F. necrophorum* subsp. *necrophorum* and subsp. *funduliforme* | 5.28 and 5.60 | 5.10 and 5.03 | 0.15 | <0.01 |
| subsp. *necrophorum* (without subsp. *funduliforme*) | 4.95 | 4.92 | 0.16 | 0.79 |
| subsp. *necrophorum* (with or without subsp. *funduliforme*) | 5.34 | 4.89 | 0.11 | <0.01 |
| subsp. *funduliforme* (without subsp. *necrophorum*) | 5.19 | 4.78 | 0.12 | <0.01 |
| subsp. *funduliforme* (with or without subsp. *necrophorum*) | 5.36 | 4.82 | 0.11 | <0.01 |
| *T. pyogenes* | 5.00 | 4.66 | 0.16 | 0.02 |
| *S. enterica* | ND | 4.91 | NA[b] | NA |
| *F. necrophorum* + *T. pyogenes* | 5.67 + 5.43 | 5.02 + 4.65 | 0.16 | <0.01 |
| *F. necrophorum* + *S. enterica* | ND | 4.67 + 4.81 | NA | NA |
| *F. necrophorum* + *T. pyogenes* + *S. enterica* | ND | 4.59 + 4.59 + 4.57 | NA | NA |

[a]ND, not detected.
[b]NA, not applicable.

**TABLE 7**  Liver abscesses of feedlot cattle that are negative for both subspecies of *Fusobacterium necrophorum* and positive for *Trueperella pyogenes* and/or *Salmonella enterica* based on 4-plex quantitative PCR (qPCR) and culture methods[b]

| Bacterial species | Prevalence and concentration | No. of samples negative or positive (%) | | | P value (qPCR vs. culture method |
|---|---|---|---|---|---|
| | | qPCR | Culture method[a] | qPCR and culture method | |
| No. of liver abscesses | | 384 | 384 | 384 | NA[c] |
| Negative for *F. necrophorum* subsp. *necrophorum* and *funduliforme* | Prevalence | 54 (14.1) | 33 (8.6) | 23 (6.0) | 0.17 |
| Positive for *T. pyogenes* | Prevalence | 1/54 (1.9) | 1/33 (3.0) | 0 | 0.85 |
| | Log$_{10}$ CFU/g | 5.32 | NA | NA | NA |
| Positive for *S. enterica* | Prevalence | 0 | 4/33 (12.1) | 2/23 (8.7) | 0.20 |
| | Log$_{10}$ CFU/g | ND | NA | NA | NA |
| Positive for *T. pyogenes* + *S. enterica* | Prevalence | 0 | 0 | 0 | NA |
| | Log$_{10}$ CFU/g | ND | NA | NA | NA |
| Negative for *F. necrophorum*, *T. pyogenes*, and *S. enterica* | Prevalence | 53 (13.8) | 28 (7.3) | 21 (5.5) | <0.01 |

[a]Includes prevalence before and after enrichment for *F. necrophorum* and *S. enterica*.
[b]ND, Not detected.
[c]NA, not applicable.

## Liver abscesses, ruminal epithelial tissues, and colonic epithelial tissues negative for *F. necrophorum*

Of the 384 liver abscess samples analyzed, 54 (14.1%) and 33 (8.6%) samples were negative for both subsp. *necrophorum* and subsp. *funduliforme* by qPCR and culture methods, respectively. Only a small number of liver abscesses (23/384; 6.0%) were negative by both detection methods, indicating the total absence of *F. necrophorum* (Table 7). Only a few of the *F. necrophorum*-negative samples were positive for *T. pyogenes* and/or *S. enterica* by qPCR or culture method. A total of 53 (13.8%), 28 (7.3%), and 21 (5.5%) liver abscesses were negative for all four pathogens by qPCR, culture, and both methods, respectively (Table 7). The difference in the liver abscess samples negative for all four pathogens between the qPCR and culture method was significant (*P* < 0.01). Of the total liver abscesses (*n* = 384) analyzed in the study, a total of 252 cattle had matched ruminal epithelial and colonic epithelial tissues (Table 8). Of the 252 cattle, 40 had ruminal epithelial tissues negative for both subspecies of *F. necrophorum* by both detection methods, and 39 of those had *F. necrophorum*. Similarly, 141 cattle were negative for the two subspecies in the colonic tissues, and 125 had *F. necrophorum* in liver abscesses. Only 25 cattle were negative for the two subspecies in both ruminal and colonic tissues, and 24 of them had *F. necrophorum* in liver abscesses.

**TABLE 8**  Cattle with ruminal epithelial tissues and/or colonic epithelial tissues negative for both subspecies of *Fusobacterium necrophorum* and positive for *F. necrophorum* in liver abscesses based on 4-plex quantitative PCR (qPCR) and culture methods

| Item | No. of samples negative (%) | | | | Liver abscesses positive for *F. necrophorum* |
|---|---|---|---|---|---|
| | qPCR | Culture method | | qPCR and culture method | |
| | | Before enrichment | Total[a] | | |
| No. of ruminal epithelial tissues | 252 | 252 | 252 | 252 | NA[b] |
| Negative for *F. necrophorum* subsp. *necrophorum* and subsp. *funduliforme* | 75 (29.8) | 116 (46.0) | 78 (31.0) | 40 (15.9) | 39 |
| No. of colonic epithelial tissues | 252 | 252 | 252 | 252 | NA |
| Negative for *F. necrophorum* subsp. *necrophorum* and subsp. *funduliforme* | 211 (83.7) | 184 (73.0) | 164 (65.1) | 141 (56.0) | 125 |
| No. of ruminal and colonic epithelial tissues | 252 | 252 | 252 | 252 | NA |
| Negative for *F. necrophorum* subsp. *necrophorum* and subsp. *funduliforme* | 61 (24.2) | 87 (34.5) | 51 (20.2) | 25 (9.9) | 24 |

[a]Total includes combined prevalence of *F. necrophorum* in samples before and after enrichment.
[b]NA, not applicable.

## DISCUSSION

The validated 4-plex qPCR is novel because the assay detects presence and provides concentrations of the four pathogens, the two subspecies of *F. necrophorum*, *T. pyogenes*, and *S. enterica*, frequently isolated from liver abscesses of cattle (12). The assay is an improvement of our previously published two qPCR assays, one to detect and differentiate the two subspecies of *F. necrophorum* and the second to detect and quantify *S. enterica*. The qPCR assay for *F. necrophorum* targeted the promoter region of the *lktA* gene, which encodes for the leukotoxin (22), and *invA* gene (34), which encodes for invasion protein gene, a critical element in the type III secretion system and essential for cellular invasion, to target *S. enterica* (33). For *T. pyogenes,* the *plo* gene, which encodes for pyolysin, an exotoxin that is cytotoxic to erythrocytes and leukocytes and a primary virulence factor, was included (30, 31). The primers and probes were designed to multiplex the assay for the detection and quantification of the four targeted genes. The qPCR assay, validated for specificity and sensitivity, was used to analyze 384 liver abscess samples, 374 ruminal epithelial tissue, and 256 colonic epithelial tissues from three different feedlot studies to determine prevalence and concentrations of the four pathogens. This is the first study that provides concentration data of all four pathogens in liver abscesses and matched ruminal and colonic epithelial tissues. The culture-based data of the three individual feedlot studies on the prevalence of the four pathogens have been described (23, 41, 42). In this study, we included the pooled data from three studies of culture method of detection of the pathogens before enrichment and total prevalence, which includes both pre-enrichment and post-enrichment combined.

Of the four pathogens associated with liver abscesses, there is substantial evidence and a near unanimity that *F. necrophorum* is the most frequently isolated and therefore is accepted as the primary etiologic agent (8). The frequency of isolation of the species has ranged from 85% to 100% of liver abscesses that have been cultured (9–12, 43). Of the two subspecies, the subsp. *necrophorum* is much more frequently isolated from liver abscesses than the subsp. *funduliforme* (21, 43, 44). The qPCR assay of the 384 liver abscess samples indicated a prevalence of 85.9% at the species level (either of the two subspecies) and 69% and 38.3% for the subsp. *necrophorum* and subsp. *funduliforme*, respectively. Almost 50% of samples that were positive for subsp. *necrophorum* did not contain the subsp. *funduliforme,* and only less than 20% contained subsp. *funduliforme* alone. The prevalence by the qPCR assay at the species and subspecies levels was similar to the culture method of samples before enrichment. Inclusion of the post-enrichment data provided a slightly higher prevalence because enrichment enhances the probability of isolations from samples. In a summary of 846 liver abscess samples analyzed by the culture method over a 25-year period, the prevalence of subsp. *necrophorum* was 89% compared to 23.5% for subsp. *funduliforme* (8). The two subspecies of *F. necrophorum* differ in a number of phenotypic characteristics, including virulence (19). The difference in virulence is related to the difference in production of virulence factors between the two subspecies, particularly in the secretion of leukotoxin, considered as the critical factor involved in the survival and proliferation in infected tissues (45–48).

In the gut, the prevalence of both subspecies of *F. necrophorum* in ruminal epithelial tissue was significantly higher than in the colonic epithelium, reaffirming the importance of the ruminal ecosystem as the source of the pathogen reaching the liver. The frequency of isolation of *F. necrophorum* from ruminal epithelial tissue, compared to qPCR-based prevalence, was much lower than from liver abscesses, likely because bacterial population in ruminal epithelium is much denser and more diverse than in liver abscesses (49). In contrast, the culture method indicated higher prevalence in the colonic epithelial tissue than the qPCR-based prevalence. It is possible that the bacterial population in the mucosal lining with a monolayer of epithelial cells is not as dense and diverse as in ruminal epithelium, thus reducing competition when plated on nonselective medium like blood agar. The reason for the difference between the two subspecies in epithelial tissues likely reflects differences in prevalence in ruminal contents, which is the logical source for the organisms in gut tissues. In a study that analyzed 345 ruminal content

samples collected in an abattoir immediately after slaughter, the subsp. *funduliforme* was detected in almost all samples (98% to 100%), and the subsp. *necrophorum* was prevalent in about 29% of samples, but the concentrations of the two subspecies were similar (22).

The concentrations of both subspecies of *F. necrophorum* were in the range of 7 to 7.5 $\log_{10}$ per gram of purulent material, either alone or together. The concentrations of both subspecies were similar when present together, which suggests that both contributed to the infection. In liver abscesses where only one subspecies was present, there was no difference in concentrations, suggesting that subsp. *funduliforme*, a less virulent subspecies, was able to cause infection in about 17% of liver abscesses without the aid of subsp. *necrophorum*. The concentrations of *F. necrophorum* in ruminal and colonic epithelial tissues were two $\log_{10}$ units lower than the concentration in liver abscesses. The concentrations in colonic epithelial tissue were consistently lower than in the ruminal epithelium. This is the first report that provides *F. necrophorum* concentration data in colonic epithelial tissues and contributes to the evidence that the hindgut could be another source of the pathogens that reach the liver.

*Trueperella pyogenes,* a Gram-positive, rod-shaped, and facultatively anaerobic organism, is frequently isolated as a single or mixed culture from a variety of pyogenic infections in cattle often associated with *F. necrophorum* (50–53). The organism is a commensal on the skin and epithelia of the upper respiratory, digestive, and reproductive tracts of cattle (51, 52). In liver abscesses, *T. pyogenes* is the second most frequently isolated pathogen next to *F. necrophorum* (12, 43, 46), and a similar association has been reported in foot rot and abscesses in cattle (48) and metritis in dairy cows (54). The association is likely because of the nutritional and pathogenic synergy between the two species (19). The qPCR assay detected higher prevalence of *T. pyogenes* than the culture method, and the difference may simply be because no specific medium is available for enrichment and selective isolation from tissue samples with mixed infections. The prevalence of *T. pyogenes* reported in the current study (29.2%) is similar to reports of isolations from liver abscesses of crossbred beef cattle (21, 44, 55), calf-fed Holsteins (44), cull-beef cattle (43), and beef-on-dairy cattle (23). There is evidence of variation in prevalence of *T. pyogenes* based on geographic location (43). The higher prevalence of *T. pyogenes* in ruminal epithelial tissue than in colonic epithelial tissue confirms the previous observation that the likely dominant source is the ruminal epithelial tissue (56). The concentrations of *T. pyogenes* in liver abscesses were about two logs lower than the concentrations of the two subspecies of *F. necrophorum*. The lower concentration does support the role as a secondary pathogen but does raise the question of the extent of its contributions to the infection. Even in 23 out of 384 liver abscess samples that were totally negative for *F. necrophorum* by both culture and PCR-based methods, the concentration of *T. pyogenes* averaged 5.32 $\log_{10}$ CFU/g, and the lack of higher concentration indicates that it is unlikely to be a primary etiologic agent.

None of the 384 liver abscess samples analyzed by the qPCR assay had quantifiable concentrations of *S. enterica,* suggesting that, if prevalent, the concentrations were less than $\log_{10}$ 3 CFU/g, which is the LLOD of the assay. The absence of quantifiable concentration does raise the question of whether *S. enterica* is an etiologic agent or even contributes to the infection. In humans, *S. enterica*, particularly certain serotypes (Typhi, Paratyphi, and Enteritidis), have been shown to cause pyogenic liver abscesses in pure culture (57–59). The prevalence of *S. enterica* in liver abscesses is much more variable than *F. necrophorum* or *T. pyogenes*. Only four studies have reported isolations of *S. enterica* from naturally occurring liver abscesses in cattle (23, 43, 44, 60). Generally, the prevalence of *Salmonella* in liver abscesses has ranged from 20% to 30%, and similar to *T. pyogenes*, there is indication that the prevalence may depend on the geographic region of the feedlots (43). Similar to liver abscesses, none of the ruminal epithelial tissue had quantifiable concentration. However, a small proportion of colonic epithelial tissue had quantifiable concentrations. The higher prevalence of *S. enterica* in colonic epithelial tissue than in ruminal epithelial tissue suggests that the colon, more likely than the rumen, may be the source of *S. enterica* reaching the liver.

A couple of significant findings of this study were as follows: (i) liver abscesses that contained either or both subspecies of *F. necrophorum* had concentrations in the range of $\log_{10}$ 7.0, indicating robust growth and contributing to the infections, and (ii) only a small proportion of liver abscesses was negative for *F. necrophorum* by either method (14.1% by qPCR and 8.6% by culture) or both methods (6%). This affirms the long-held observation that *F. necrophorum* is the primary causative agent of liver abscesses. Only a few of the *F. necrophorum*-negative samples contained the other two pathogens, *T. pyogenes* and *S. enterica*, and the concentrations of these two pathogens did not increase in the absence of *F. necrophorum,* which suggests that they were unlikely to contribute to the development of abscesses. A total of 21 of 384 abscesses (6%) were caused by bacteria other than the three frequently isolated pathogens, i.e., *F. necrophorum*, *T. pyogenes*, and *S. enterica*. Interestingly, 25 cattle were negative for both subspecies of *F. necrophorum* by qPCR and culture methods in rumen and colonic epithelial tissues, yet 24 had liver abscesses that contained *F. necrophorum*, suggesting that the source of the pathogen is likely to be other than the gastrointestinal tract. Besides the gut, *F. necrophorum* is prevalent in other regions of the bovine, such as the oral cavity and the respiratory and reproductive tracts (61–65). *Fusobacterium necrophorum* causes foot abscesses or foot rot, and the source is believed to be the soil (48). Thus far, no evidence is available to suggest that the source of liver abscess pathogens is anything other than the gut.

## Conclusions

The novel qPCR assay provided both prevalence and concentrations of the four frequently isolated pathogens from liver abscesses: *F. necrophorum* subsp. *necrophorum*, *F. necrophorum* subsp. *funduliforme*, *T. pyogenes*, and *S. enterica*. The prevalence and concentration data for *F. necrophorum*, particularly the subsp. *necrophorum*, affirm its primary role in the cause of liver abscesses. Only a small number of liver abscesses were caused primarily by the subsp. *funduliforme*. The concentration data for *T. pyogenes* suggest that it is unlikely to be the primary etiologic agent. Although *S. enterica* was prevalent in liver abscesses, none of them had concentrations >$10^3$ CFU per gram, which suggests that the species is not contributing to the development of liver abscesses.

## ACKNOWLEDGMENTS

The manuscript is contribution no. 25-234-J from the Kansas Agric. Exp. Station, Manhattan. The study was supported in part from funds provided by the North Central 1202 Committee on Enteric Diseases of Food Animals and by the Foundation for Food and Agricultural Research-International Consortium for Antimicrobial Stewardship in Agriculture.

T.G.N. and J.B. conceptualized the study; T.G.N., R.G.A., D.R.W., T.E.L., and K.L.S. designed the feedlot studies; R.A.W., K.N.S., D.R.W., T.E.L., and K.L.S. collected samples; T.G.N., J.B., X.S., L.A.G., M.A., A.D., H.S., and R.G.A. were responsible for the methodology; M.A., X.S., L.A.G., A.D., H.S., and R.G.A. processed and analyzed the samples and captured the data; R.A.W. analyzed the data; and M.A. and T.G.N. drafted the initial manuscript. All authors read the manuscript, contributed to editing, and approved the final version submitted.

## AUTHOR AFFILIATIONS

[1]Department of Diagnostic Medicine and Pathobiology, Kansas State University, Manhattan, Kansas, USA
[2]Department of Clinical Sciences, Kansas State University, Manhattan, Kansas, USA
[3]Department of Animal and Food Sciences, Texas Tech University, Lubbock, Texas, USA
[4]Department of Agricultural Sciences, West Texas A&M University, Canyon, Texas, USA
[5]Veterinary Diagnostic Laboratory, Kansas State University, Manhattan, Kansas, USA

## AUTHOR ORCIDs

Mina Abbasi http://orcid.org/0000-0003-2039-4134
Alyssa Deters https://orcid.org/0009-0004-2885-0688
Harith Salih http://orcid.org/0009-0001-6375-7938
Reese A. Wilson http://orcid.org/0009-0006-8510-4467
Kasi N. Schneid http://orcid.org/0000-0003-0086-3028
Xiaorong Shi http://orcid.org/0000-0002-1800-8719
Leigh Ann George http://orcid.org/0000-0001-9335-450X
Jianfa Bai http://orcid.org/0000-0002-0021-3036
Raghavendra G. Amachawadi http://orcid.org/0000-0001-9689-1124
Dale R. Woerner http://orcid.org/0000-0002-2063-4807
Kendall L. Samuelson http://orcid.org/0000-0001-7116-0173
Ty E. Lawrence http://orcid.org/0000-0001-7088-5455
T. G. Nagaraja http://orcid.org/0000-0002-0899-878X

## AUTHOR CONTRIBUTIONS

Mina Abbasi, Data curation, Formal analysis, Investigation, Methodology, Writing – original draft | Alyssa Deters, Investigation, Methodology, Writing – review and editing | Harith Salih, Investigation, Writing – review and editing | Reese A. Wilson, Data curation, Formal analysis, Investigation, Writing – review and editing | Kasi N. Schneid, Investigation, Writing – review and editing | Xiaorong Shi, Investigation, Methodology, Validation, Writing – review and editing | Leigh Ann George, Investigation, Methodology, Writing – review and editing | Jianfa Bai, Conceptualization, Methodology, Validation, Writing – review and editing | Raghavendra G. Amachawadi, Investigation, Writing – review and editing | Dale R. Woerner, Investigation, Resources, Writing – review and editing | Kendall L. Samuelson, Investigation, Resources, Writing – review and editing | Ty E. Lawrence, Resources, Writing – review and editing | T. G. Nagaraja, Conceptualization, Data curation, Funding acquisition, Investigation, Methodology, Project administration, Resources, Supervision, Writing – review and editing

## DATA AVAILBILITY

All the data generated are presented in tables in the manuscript

## ETHICS APPROVAL

All the animal handling and management procedures in this study were approved by the relevant Institutional Animal Care and Use Committee (IACUC). In the first feedlot study conducted in 2022, the samples were collected at an abattoir from slaughtered cattle; therefore, no IACUC approval was required. The second feedlot study was conducted at Hy-Plains Feedyard, LLC located in Montezuma, Kansas, and all procedures in the care and use of animals were approved by the Veterinary Research and Consulting Services, LLC, IACUC (no. 1012). The procedures used in the third feedlot study were approved by the IACUC at West Texas A&M University, Canyon, TX, USA (no. 2022.07.002).

## ADDITIONAL FILES

The following material is available online.

Open Peer Review

**PEER REVIEW HISTORY (review-history.pdf).** An accounting of the reviewer comments and feedback.

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
