## [Reviewer comments · Microbiology Spectrum]

Microbiology Spectrum

Quantification of Major Bacterial Pathogens in Liver Abscesses and Matched Ruminal and Colonic Epithelial Tissues of Feedlot Cattle

Mina Abbasi, Alyssa Deters, Harith Salih, Reese Wilson, Kasi Schneid, Xiaorong Shi, Leigh Ann George, Jianfa Bai, Raghavendra Amachawadi, Dale Woerner, Kendall Samuelson, Ty Lawrence, and T. Nagaraja

Corresponding Author(s): T. Nagaraja, Kansas State University

Review Timeline:

Submission Date:	June 18, 2025
Editorial Decision:	July 21, 2025
Revision Received:	July 25, 2025
Accepted:	July 26, 2025

Editor: Artem Rogovsky

Reviewer(s): Disclosure of reviewer identity is with reference to reviewer comments included in decision letter(s). The following individuals involved in review of your submission have agreed to reveal their identity: Robert J Gruninger (Reviewer #3)

Transaction Report:

DOI: <https://doi.org/10.1128/spectrum.01888-25>

Re: Spectrum01888-25 (**Quantification of Major Bacterial Pathogens in Liver Abscesses and Matched Ruminant and Colonic Epithelial Tissues of Feedlot Cattle**)

Dear Dr. T. G. Nagaraja:

Thank you for the privilege of reviewing your work. Below you will find my comments, instructions from the Spectrum editorial office, and the reviewer comments.

Revision Guidelines

Sincerely,
Artem Rogovsky
Editor
Microbiology Spectrum

Reviewer #1 (Comments for the Author):

In this manuscript, Abbasi et al. developed a 4-plex quantitative PCR (qPCR) assay to simultaneously detect and quantify four major bacterial pathogens-Fusobacterium necrophorum subsp. necrophorum (FNN), F. necrophorum subsp. funduliforme (FNF), Trueperella pyogenes (TP), and Salmonella enterica (SE)-in liver abscesses (LAs) from cattle, as well as in matched ruminal and colonic epithelial tissues. Their goal was to test the hypothesis that species abundance in purulent material could be

indicative of the pathogen's role in disease.

By analyzing a large number of samples, the authors found that FNN and FNF were most frequently detected by the qPCR assay. The qPCR results were consistent with those obtained by culture-based methods, thereby validating the qPCR approach. Interestingly, only 16.9% of LA samples were positive for FNF in the absence of FNN. While 29.2% of samples were positive for TP, SE was not detected in any of the samples using the qPCR assay-despite previous studies reporting its prevalence in LA cases. Notably, this study is the first to report concentration data for TP and SE in liver abscess samples.

This novel 4-plex qPCR assay provides both prevalence and concentration data for four commonly isolated LA pathogens, and the findings reinforce the central role of FNN in the pathogenesis of liver abscesses. The results suggest that TP and SE are unlikely to be primary etiologic agents.

The manuscript is well-written, clearly presented, and easy to follow. The data are convincing and support the authors' conclusions.

One Major Question:

The authors mention that *S. enterica* was previously found to be prevalent in liver abscesses, yet none of the samples in this study had detectable SE concentrations using the qPCR assay. Could the authors explain why SE was not detected in any samples? Is this due to assay sensitivity, sample variability, or differences in the study population?

Minor Comments:

- Line 42: The sentence "Although SE was prevalent, none of them was quantifiable" is somewhat unclear. Consider revising to: "Although SE has been reported as prevalent in prior studies, it was not quantifiable in this study."
- Line 46: The name *Fusobacterium necrophorum* is missing an "F" at the beginning. Please correct.
- Line 76: Typo: "Pseudomonodota" should be corrected to "Pseudomonadota."
- Line 101: For clarity, add "*Fusobacterium*" in front of "varium," i.e., "*Fusobacterium varium*."
- Line 107: Instead of "the two subspecies of *Fusobacterium*," write "the two subspecies of *Fusobacterium necrophorum*."
- Line 113: Remove the comma in "*F. necrophorum*, subsp. *necrophorum*."
- Line 438: Citation error - "(O'Hara et al., 20220)" should be corrected to "(O'Hara et al., 2022)."

Reviewer #3 (Comments for the Author):

The authors describe the development of a 4-plex quantitative PCR (qPCR) assay to determine the prevalence and concentrations of the pathogens in LA (n=384) and matched ruminal (RT; n=374) and colonic epithelial (CT; n=256) tissues, collected at slaughter, from three feedlot studies. This is a great study and the fact that there are so many matched tissue samples with LA samples is very unique. The manuscript is well written and the results would be of interest to researchers in the field and will provide a validated method that can be adopted by others. The work is rigorous and I don't have any major concerns with the methodology. I have some minor comments that should be easily addressed.

L46 - missing F in *Fusobacterium*

L202 - you mention Correlation coefficients and PCR amplification efficiencies were calculated but I can't seem to find where these are reported. This information would fit in with the results section where lower limits of detection are reported.

L207 - please indicate storage conditions of samples. The 2002 samples will be 23 years old now. Do you think this may impact the ability to detect the microbes of interest from these samples?

L286 - I am a bit confused about where the culture based prevalence data is from. I can't seem to find where in the methods the isolation procedures are described. I can find the information that I believe is related to the standard strains used for the qPCR development but not for the isolation from the experimental samples. - Once I reached the discussion I found the information on where the culture data is from but I think this needs to be clearly indicated earlier in the paper

L397 - missing year in Bai et al reference

L438 - Error in O'Hara et al 2022 reference

L422 - If the isolation prevalence information is from previously published studies you should be sure you reference those in the text after you refer to it.

L450 - 'both contributed to the infection'

L486 - Was the *S. enterica* strain used to develop the assay from a LA sample? Are there other species/serotypes (*S. lubbock*?) that have been isolated from LA that could have been missed?

Table 4. Seems to be missing the statement "Total includes prevalence before and after enrichment for...." At the bottom of the table

AUTHORS' RESPONSE TO REVIEWERS AND EDITORS' COMMENTS

Reviewer #1 (Comments for the Author):

In this manuscript, Abbasi et al. developed a 4-plex quantitative PCR (qPCR) assay to simultaneously detect and quantify four major bacterial pathogens—*Fusobacterium necrophorum* subsp. *necrophorum* (FNN), *F. necrophorum* subsp. *funduliforme* (FNF), *Trueperella pyogenes* (TP), and *Salmonella enterica* (SE)—in liver abscesses (LAs) from cattle, as well as in matched ruminal and colonic epithelial tissues. Their goal was to test the hypothesis that species abundance in purulent material could be indicative of the pathogen's role in disease.

By analyzing a large number of samples, the authors found that FNN and FNF were most frequently detected by the qPCR assay. The qPCR results were consistent with those obtained by culture-based methods, thereby validating the qPCR approach. Interestingly, only 16.9% of LA samples were positive for FNF in the absence of FNN. While 29.2% of samples were positive for TP, SE was not detected in any of the samples using the qPCR assay—despite previous studies reporting its prevalence in LA cases. Notably, this study is the first to report concentration data for TP and SE in liver abscess samples.

This novel 4-plex qPCR assay provides both prevalence and concentration data for four commonly isolated LA pathogens, and the findings reinforce the central role of FNN in the pathogenesis of liver abscesses. The results suggest that TP and SE are unlikely to be primary etiologic agents.

The manuscript is well-written, clearly presented, and easy to follow. The data are convincing and support the authors' conclusions.

One Major Question:

The authors mention that *S. enterica* was previously found to be prevalent in liver abscesses, yet none of the samples in this study had detectable SE concentrations using the qPCR assay. Could the authors explain why SE was not detected in any samples? Is this due to assay sensitivity, sample variability, or differences in the study population?

RESPONSE: Thank you for raising this question. The qPCR assay was done with samples before enrichment only because we did store the post-enrichment samples in the previous studies. Our qPCR assay has a lower limit of detection of approximately 10^3 CFU/g. In previous studies and in the current report (Table 2), we have reported prevalence of 3.1% in samples before enrichment and 19.8% in samples after enrichment. A few samples (3.1%) was positive by culture method because with streaking a loopful of culture, the detection requires only one *Salmonella* colony (theoretically, the detection limit of direct plating is 10^2 CFU per g) p consider the sample as psoitive . We did not save post-enrichment samples from previous studies. If we had those, qPCR would have identified a few as positive samples. We have expanded the discussion to address this issue. (Please see lines 422-424)

Minor Comments:

- Line 42: The sentence "Although SE was prevalent, none of them was quantifiable" is somewhat unclear. Consider revising to:
"Although SE has been reported as prevalent in prior studies, it was not quantifiable in this study."

RESPONSE: We have revised to provide clarity (Please see lines 39-40)

- Line 46: The name *Fusobacterium necrophorum* is missing an "F" at the beginning. Please correct.

RESPONSE: Corrected (Please see line 44)

- Line 76: Typo: "Pseudomonodota" should be corrected to "Pseudomonadota."

RESPONSE: Corrected (Please see line 74)

- Line 101: For clarity, add "*Fusobacterium*" in front of "varium," i.e., "*Fusobacterium varium*."

RESPONSE: Corrected (Please see line 98)

- Line 107: Instead of "the two subspecies of *Fusobacterium*," write "the two subspecies of *Fusobacterium necrophorum*."

RESPONSE: Corrected (Please see line 105)

- Line 113: Remove the comma in "*F. necrophorum*, subsp. *necrophorum*."

RESPONSE: Corrected (Please see line 111)

- Line 438: Citation error - "(O'Hara et al., 20220)" should be corrected to "(O'Hara et al., 2022)."

RESPONSE: Corrected (Line 453)

Reviewer #3 (Comments for the Author):

The authors describe the development of a 4-plex quantitative PCR (qPCR) assay to determine the prevalence and concentrations of the pathogens in LA (n=384) and matched ruminal (RT; n=374) and colonic epithelial (CT; n=256) tissues, collected at slaughter, from three feedlot studies. This is a great study and the fact that there are so many matched tissue samples with LA samples is very unique. The manuscript is well written and the results would be of interest to researchers in the field and will provide a validated method that can be adopted by others. The work is rigorous and I don't have any major concerns with the methodology. I have some minor comments that should be easily addressed.

L46 - missing F in *Fusobacterium*

RESPONSE: Corrected (Please see line 44)

L202 - you mention Correlation coefficients and PCR amplification efficiencies were calculated

but I can't seem to find where these are reported. This information would fit in with the results section where lower limits of detection are reported.

Response: We have complied with the suggestion (Please see lines 278-283)

L207 - please indicate storage conditions of samples. The 2002 samples will be 23 years old now. Do you think this may impact the ability to detect the microbes of interest from these samples?

Response: Thank you for pointing this out. The year “2002” was a typographical error and it should have been “2022”. This is corrected. All samples were collected in 2022 or 2023 and stored at –80 °C until analyzed (Please see line 206)

L286 - I am a bit confused about where the culture-based prevalence data is from. I can't seem to find where in the methods the isolation procedures are described. I can find the information that I believe is related to the standard strains used for the qPCR development but not for the isolation from the experimental samples. - Once I reached the discussion I found the information on where the culture data is from but I think this needs to be clearly indicated earlier in the paper.

Response: We are sorry for creating the confusion. The culture data were pooled from three feedlot studies. We have added a couple of sentences to provide clarity on the culture data and have provided references (Abbasi et al., 2025; Harith et al., 2023; Schneid et al., 2024) (Please see lines 235-241)

L397 - missing year in Bai et al reference

Response: Corrected. “Bai et al., 2018.” (Line 410)

L438 - Error in O'Hara et al 2022 reference

Response: Corrected (Line 453)

L422 - If the isolation prevalence information is from previously published studies you should be sure you reference those in the text after you refer to it.

Response: We appreciate the reviewer’s comment. Again, the data included in this publication were pooled from three individual feedlot studies (Salih et al., 2023; Abbasi et al., 2025; Schneid et al., 2025) (Please see lines 235-241)

L450 - 'both contributed to the infection'

Response: Revised for clarity. (Please see line 464)

L486 - Was the *S. enterica* strain used to develop the assay from a LA sample? Are there other species/serotypes (*S. lubbock*?) that have been isolated from LA that could have been missed?

Response: The *S. enterica* strain (2016-13-36) used in the assay development was isolated from a liver abscess samples (Lines 129-133). In addition, specificity testing included 66 *S. enterica* strains representing a wide range of serotypes, including S. Lubbock, S. Montevideo, S. Anatum, and S. Kentucky (Lines 157-166). These were used to ensure the assay's broad applicability and minimize the risk of missing genetically diverse serotypes.

Table 4. Seems to be missing the statement "¹Total includes prevalence before and after enrichment for..." At the bottom of the table

Response: Corrected. The footnote is added to the Table 4.

Re: Spectrum01888-25R1 (**Quantification of Major Bacterial Pathogens in Liver Abscesses and Matched Ruminal and Colonic Epithelial Tissues of Feedlot Cattle**)

Dear Dr. T. G. Nagaraja:

Your manuscript has been accepted, and I am forwarding it to the ASM production staff for publication. Your paper will first be checked to make sure all elements meet the technical requirements. ASM staff will contact you if anything needs to be revised before copyediting and production can begin. Otherwise, you will be notified when your proofs are ready to be viewed.

Sincerely,
Artem Rogovsky
Editor
Microbiology Spectrum